# *Drosophila* larvae form appetitive and aversive associative memory in response to thermal conditioning

**Nikolaos T. Polizos**[1,2], **Stephanie Dancausse**[1,2], **Consuelo Rios**[1,2], **Mason Klein**[1,2]*

**1** Department of Biology, University of Miami, Coral Gables, Florida, United States of America, **2** Department of Physics University of Miami, Coral Gables, Florida, United States of America

\* klein@miami.edu

**Data Availability Statement:** All larva trajectory files and analysis scripts are available from the Harvard Dataverse database (https://doi.org/10.7910/DVN/GK8G8D).

## Abstract

Organisms have evolved the ability to detect, process, and respond to many different surrounding stimuli in order to successfully navigate their environments. Sensory experiences can also be stored and referenced in the form of memory. The *Drosophila* larva is a simple model organism that can store associative memories during classical conditioning, and is well-suited for studying learning and memory at a fundamental level. Much progress has been made in understanding larval learning behavior and the associated neural circuitry for olfactory conditioning, but other sensory systems are relatively unexplored. Here, we investigate memory formation in larvae treated with a temperature-based associative conditioning protocol, pairing normally neutral temperatures with appetitive (fructose, FRU) or aversive (salt, NaCl) stimuli. We test associative memory using thermal gradient geometries, and quantify navigation strength towards or away from conditioned temperatures. We find that larvae demonstrate short-term associative learning. They navigate towards warmer or cooler temperatures paired with FRU, and away from warmer or cooler temperatures paired with NaCl. These results, especially when combined with future investigations of thermal memory circuitry in larvae, should provide broader insight into how sensory stimuli are encoded and retrieved in insects and more complex systems.

## Introduction

### General background

Animal behavior relies on sensory input. When sensory information is sent to the brain it can sometimes be encoded in the form of memory and later retrieved. Through memory, previous sensory experiences can shape future behavior. It is common for organisms to anticipate an outcome based on external and unrelated stimuli, a process that is referred to as associative memory and learning. Associative memory is a complex behavior that is important for survival and fitness for many organisms. Associative memory is ubiquitous across taxa, and is found in both invertebrates like *Aplysia* [1] and vertebrates like *Danio rerio* [2].

In the laboratory, the ability for an organism to form associative memory can be tested by implementing a classical conditioning paradigm. During classical conditioning, two stimuli

**Funding:** MK 2144385 National Science Foundation https://www.nsf.gov/ The funders played no role.

**Competing interests:** The authors have declared that no competing interests exist.

are used, and the study organism learns to form an association between them. The first stimulus normally yields a strong response from the organism when applied alone (unconditioned stimulus, US). The second stimulus is normally neutral to the organism when applied alone, and elicits no response until after the protocol is completed (conditioned stimulus, CS).

Achieving a comprehensive understanding of associative memory that includes both behavior and the underlying neural circuitry is a difficult and important task in neuroscience. Vertebrate models can be more challenging to understand due to the size of their brains (humans have $\sim$ 90 billion neurons, mice have $\sim$ 70 million). Using smaller animals like insects offers important advantages due to their more tractable nervous systems. The carpenter ant (*Camponotus aethiops*) and the honeybee (*Apis mellifera*) are able to form associations between otherwise neutral odors (CS) and painful temperature stimuli (US) [3, 4]. The *Drosophila melanogaster* larva ($< 10,000$ neurons) is an even simpler insect model. A slow-moving crawler with a transparent body, the larva is well-suited for detailed behavior analysis and *in vivo* neural imaging, and has also been shown to engage in associative learning. Here we use *Drosophila* larvae to establish a learning and memory assay that pairs gustatory stimuli (US) with normally neutral temperatures (CS) in an effort to elicit the formation of thermal associative memory.

## Larval learning and memory

The *Drosophila* larva does form associative memory. Some learning behaviors have been characterized in *Drosophila* larvae, although nearly every associative memory protocol has relied on odor as the conditioned stimulus. Typically in olfactory-based associative learning experiments, a reward (e.g., sugar) or punishment (e.g., electric shock or quinine) is paired with a neutral odorant [5–11].

At the neural circuit level, it has been established that the brain region responsible for learning and memory formation in insects is the mushroom body (MB) [12–25]. A complete connectome of the larval brain [26] as well as the specific neural circuit responsible for olfactory memory [27] have been recently mapped. A model for the proposed functions of the different neuronal populations within the olfactory learning circuit has also been proposed [28]. It remains unknown whether this circuitry is also involved in the associative learning for non-olfactory stimuli. Research in adult flies provides evidence of some shared circuitry between olfactory and visual memory [29], but a separate MB circuit could be responsible for learning involving other stimuli.

The field has developed extensive knowledge of the larval olfactory system [10, 30–37]. Therefore, there is value in developing new learning and memory protocols that include stimuli besides odorants to expand the field. On a technical level, airborn chemical concentrations can be difficult to control and measure precisely, and experimental devices may require complicated systems for delivering, cleaning, and purging odorants [38]. Light has also been used as both a CS and US in larval learning and memory studies [7, 9, 39, 40], although larvae have an innate aversion to light [41] that can complicate learning paradigms. Investigating stimuli besides odorants and light in learning and memory experiments is warranted, and in particular, the use of thermal stimuli is lacking in the current literature. Associative memory is inherently multi-sensory, so studies that focus on other stimuli should be conducted to achieve a broader understanding of memory formation and the resulting behavioral responses.

## Larval thermal sensing and response

In the present work, we seek to demonstrate the viability of temperature as a robust stimulus for fly larva conditioning experiments. Temperature is of vital importance to nearly every animal, especially small, slow ectotherms like the *Drosophila* larva. Two previously established

aspects of thermal behavioral response in larvae are especially important for developing associative memory protocols: (1) Larvae are extremely sensitive to changes in temperature (both warming and cooling) and can thus robustly navigate thermal environments [42, 43]; but (2) there is a range of temperatures where larvae do not normally exhibit thermotaxis [44], so temperatures within that range can be freely used as the conditioned stimulus (CS).

The cellular and molecular mechanisms underlying non-nociceptive temperature sensing in *Drosophila* larvae are reasonably well understood. Larvae sense changes in the temperature of their environment using three cool-sensing neurons located on each side of their heads [42] as well as two warm-sensing neurons of opposite valence [45], both housed in the dorsal organ ganglion. The three cool-sensing cells are extremely sensitive to very small changes in temperature [42, 43], responding to changes as small as a few thousandths of a ˚C per second. Three sub-types of ionotropic receptors expressed by the cool-sensing cells, Ir21a, Ir25a, and Ir93a, are all necessary for the temperature sensitivity of these neurons [46, 47]. In terms of learning and memory experiments, nociceptive temperatures have been used as an unconditioned stimulus (US) in adult flies [48] and larvae [49], with subsequent testing in binary choice assays. We deploy non-nociceptive temperatures in the present work. There is a temperature range between cold and warm avoidance regimes (approximately 22–28˚C) where larvae do not normally exhibit thermal preference by moving either up or down a thermal gradient [44]. Individual cooling and warming sensory neurons themselves do remain sensitive to temperature changes in this range, but a cross-inhibition mechanism effectively cancels their competing signals [45]. In this paper we take advantage of the neutral temperature range and use temperature as a CS in learning protocols, and show that larvae are indeed able to form associative memories with temperature-based conditioning and navigate accordingly.

## Materials and methods

### Fly strains and husbandry

All *Drosophila* stocks were raised at room temperature ($\sim$ 24–25˚C) in test tubes (Genesee) with a cornmeal and molasses food base. Second instar larvae were collected with cages (Genesee) where eggs were laid on Petri dishes containing a 1.5% agar-grape juice mixture topped with re-hydrated inactive dry yeast ($\sim$ 0.35–0.45 g). The cages and Petri dishes were also kept at room temperature.

Two wild-type strains were used: an isogenic line of $w^{1118}$ (a gift from Sheyum Syed), which has a null mutation of the *white* gene, and Canton-S [50] (BDSC #64349). We also used the loss of function mutant $Ir25a^2$ (BDSC #41737), which lacks a required receptor for cooling and warming sensing in the dorsal organ.

### Temperature gradient platforms

Two temperature-controlled platforms were used for testing larval navigation performance post-conditioning (Fig 1). One platform establishes a linear gradient in the *x*-direction across a square arena, each side set to a specific temperature with a linear transition in between. The other platform establishes a radial gradient, where the temperatures at the center and perimeter are given different set points, establishing a linear gradient along any radius. Both customized platforms were machined from aluminum blocks. The linear gradient consisted of a top plate (30.5 cm deep in the *y*-direction, 71 cm wide in the *x*-direction, 6.4 mm thick) with a reservoir block at each end (30.5 cm deep, 10.2 cm wide, 3.8 cm thick). Agar gels for the linear gradient platform were 22×22 cm squares. The radial gradient platform consisted of a top plate (30.5 cm in diameter, 1.25 cm thick) mounted on a circular reservoir block (30.5 cm in diameter, 4.5 cm tall). The reservoir block was hollowed in order to make continuous contact (1 cm)

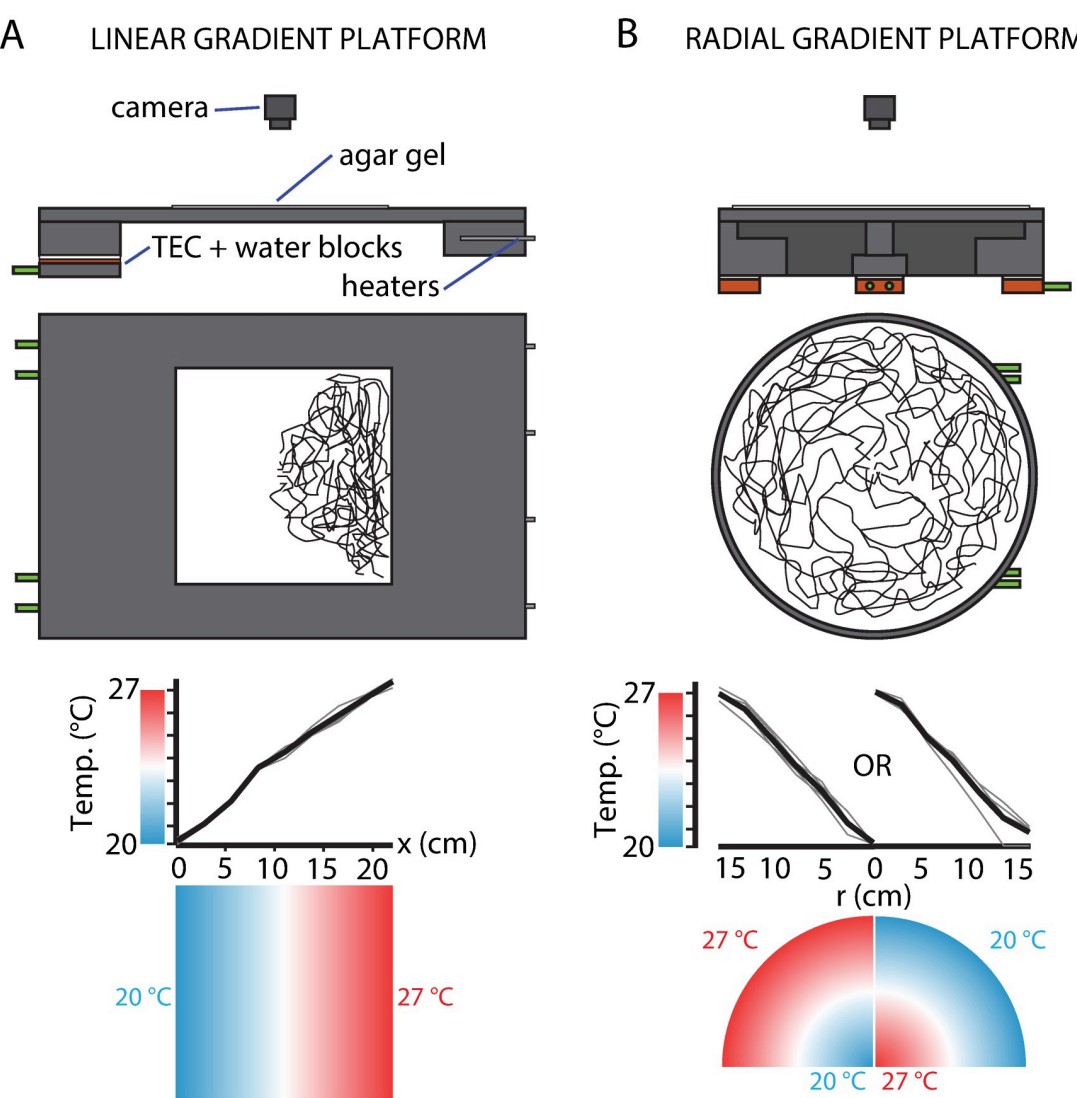

**Fig 1. Thermotaxis testing platforms.** An agar gel (white) is placed on the surface of these units. A: Top: a stable 1D linear gradient is produced by maintaining a constant cool temperature on one end and a constant warm temperature on the other (as in [42, 44] and elsewhere), using thermoelectric coolers (TECs) and resistive heaters. An agar gel (white) is placed on the surface of the unit. Black traces represent larva crawling tracks. Bottom: Temperature across the width of the crawling surface. Individual traces from repeated measurements in gray, average in black. The temperature color map is shown below. B: Top: a stable radial gradient, produced using 6 TECs under the outer section to maintain a constant temperature around the perimeter and another more powerful TEC in the center to establish the opposing temperature. An agar gel (white) is placed on the surface of the unit. Black traces represent larva crawling tracks. Bottom: Temperature along a radius, individual measurements in gray, average in black. The temperature color map is shown below, in both possible configurations. After conditioning, larvae are placed on one of these platforms and their movement is recorded with a camera. Drawings are to scale, with each gel 22 cm wide.

with the outer edge of the top plate, and to allow space for a separate central cooling unit (void 18.5 cm in diameter). The center cooling unit was mounted to a circular aluminum block (5.08 cm in diameter) joined with an aluminum cylinder (2.5 cm in diameter, 2.0 cm tall), which makes contact with the center of the top plate. Agar gels for the radial gradient platform were circular with a diameter of 30.5 cm, poured directly onto the top plate. A silicone barrier held the liquid in place as the gel cooled. The gels for both platforms were 3.5 mm thick. The gel composition was 2.5% wt./vol. agar (MilliporeSigma) and 0.75% wt./vol. charcoal (Fisher

Scientific), except for experiments with NaCl (MilliporeSigma) on the testing gel, where the composition was 1.5% wt./vol. agar, 0.75% wt./vol. charcoal, 8.7% wt./vol. NaCl (1.5 M). The agar gels were placed atop each platform and acted as crawling substrates for recording larva behavior.

The temperatures of each gradient were measured by RTDs (McMaster-Carr) or thermocouples (Physitemp), and maintained by thermoelectric coolers (TECs) (Custom Thermoelectric) driven by PID controllers and H-bridge amplifiers (Accuthermo). Copper water blocks (Custom Thermoelectric), attached to the underside of each TEC, were connected to a circulating chiller (VWR) to dissipate excess heat. Both systems allowed us to monitor and control temperature at the extremes of each gradient with high precision (0.1˚C), maintaining stable gradients throughout all experiments.

## Image acquisition

The temperature-controlled platforms were each surrounded by a square perimeter of red LEDs (620 nm) that provided dark field illumination to the arena. Using black agar gels aided image contrast. An above-mounted CCD camera (Basler) with an 8-mm lens (Computar) continuously recorded many crawling larvae simultaneously during each 10-minute experiment. Images were recorded at 15 frames per second, faster than any larval behavior of interest. After each experiment, videos were run through a customized analysis pipeline to quantify behavior.

## Video analysis

Raw image data was processed using the MAGAT Analyzer software [51], which extracts the positions and body contours of each individual larva at every frame, determines relevant locomotion parameters, and segments each trajectory into an alternating sequences of relatively straight-crawling "runs" and direction-altering "turns," akin to a modified random walk [52].

Custom analysis programs written in Matlab and Igor Pro extracted properties of each run and turn, which were used to calculate behavioral characteristics at population and individual levels. For example, the number of turns $N$ divided by the sum of run durations $T = \sum \Delta t_i$ yields the turning rate $R = N/T$, typically the most important parameter for random walks. When $R$ and other parameters vary with crawling direction (due to varying stimulus inputs), trajectories on average will exhibit directed motion, up or down a stimulus gradient.

We used the navigation index (*NI*) as a summary metric for performance on the thermal gradients. For the linear gradient this was computed as $NI_x = \langle v_x \rangle / \langle v \rangle$, the average $x$-component of velocity normalized by the average speed during runs. Equivalently, $NI_x$ is the weighted average of $\cos\theta$, where $\theta$ indicates the crawling direction ($\theta = 0$ points in the $+x$ direction). For radial gradients we instead computed $NI_r = \langle v_r \rangle / \langle v \rangle$, where $r$ is the radial direction. Both navigation indexes distill larva behavior into a dimensionless number that summarizes the efficiency of crawling towards a specific direction, and they emphasize the strategies that shape trajectory headings because they are independent of overall speed.

Only tracks that started before the 300 s mark in their corresponding experiment and lasted more than 300 s were included. This ensured that each individual animal corresponded to at most one track, avoiding any double counting. With such filtering in place, an overall population navigation index could be computed as the mean of the individual animals' *NI* values.

## Learning protocols

**Conditioning**. Prior to any recording or testing, larvae were repeatedly exposed to paired temperature (conditioned stimulus, CS) and tastant (unconditioned stimulus, US) combinations in an attempt to elicit associative memory formation (Fig 2). All conditioning was conducted

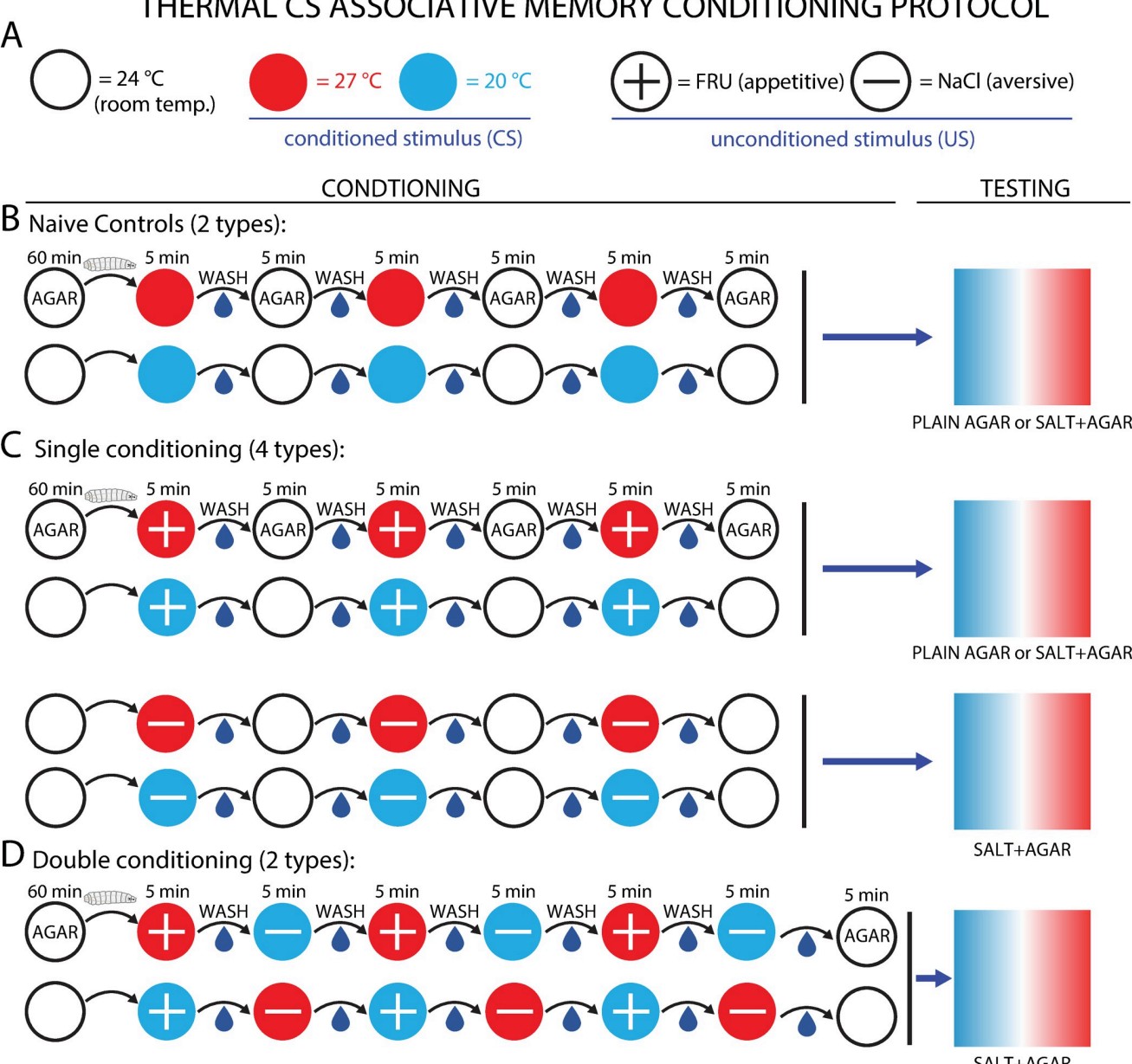

**Fig 2. Protocols for establishing associative memory with temperature and tastants.** A: Symbol legend, used here and in all figures. Circles denote the US (by color) and CS (by + or − symbol) to describe the paired conditioning used in an experiment. B: Control protocols. Naive controls were performed by using a plain agar plate (without the US tastant), but with the CS temperature still present. All larvae were washed and held on a plain agar plate for 60 min at room temperature ($\sim$ 24°C, white circles), then transferred to an agar plate held at one of the two CS temperatures, and remained there for 5 min. Larvae were then washed and transferred to another plain agar gel for an additional 5 min at room temperature ($\sim$ 24°C). This cycle was repeated as shown in the schematic. Empty white circles indicate plain agar for all protocols, and the durations from the first protocol are the same for all other protocols. After conditioning larvae were tested on a temperature gradient (Fig 1). During testing, larvae crawl on plain agar gel, or agar gel with salt at 1.5 M. C: Single conditioning protocols. Same as control, but larvae are transferred to an agar plate where the CS and US are paired, and remain there for 5 min. D: Double conditioning. Except at the beginning and end, the plain agar gel phase was replaced by a second paired gel of the opposite CS and US. Each experimental trial used 15–20 individual larvae.

in a dark room, with light for human viewing provided by a red (620 nm) LED lamp, a color that lies outside the larva's visual range.

Appetitive conditioning used fructose (FRU) as a reward, where FRU (MilliporeSigma) was included in 1.5% agar gels during the conditioning phase (2 M FRU) Gels with similar FRU concentrations have been used as the US reward in other conditioning studies with odors as the CS [8, 53]. Aversive conditioning used sodium chloride (NaCl) as a punishment, where NaCl was included in 1.5% agar gels (1.5 M NaCl). Sodium chloride at similar concentrations has been shown to be an adequately aversive US in other larval conditioning studies [7, 10, 32, 54]. We ran preliminary experiments using a bitter tastant (quinine) or electric shock as the aversive US, but did not observe strong behavior changes with quinine, and the electric current in shock experiments heated the larvae, which confounded the thermal CS component of the protocols. As a result we used NaCl as the aversive US for all experiments presented in this work.

The specific procedure for conditioning (Fig 2) was performed as follows: Second instar larvae were removed from rearing cages and rinsed in DI water to remove food and waste. They were then placed in a Petri dish containing plain agar (1.5% wt./vol.) for 60 minutes. After this starvation period, larvae were transferred to a new Petri dish for paired conditioning, where the agar gel contained either FRU or NaCl (US) and the gel was held at a fixed temperature of either 20 or 27˚C (CS). Larvae were kept in the paired conditioning gel for 5 minutes, then rinsed in DI water and moved to a new Petri dish. During single conditioning, larvae were transferred to a second agar gel absent of any US, and held at room temperature ($\sim$ 24˚C). These larvae were compared to naive control larvae that experienced the same CS during conditioning in absence of a US. During double conditioning, larvae were transferred to a second gel containing the opposite US/CS pairing from the first gel. After 5 minutes on the second dish, the larvae were rinsed again and returned to the first dish. This process was repeated twice more, such that larvae were subjected to each temperature/tastant (CS/US) pairing for three 5-minute intervals. For double conditioning protocols, larvae were finally transferred to a plain agar gel at room temperature to rest for 5 minutes.

**Testing**. After undergoing the conditioning protocol, larvae were transferred to one of the two testing platforms (Fig 1) and their crawling behavior was recorded by an overhead camera for 10 minutes. In appetitive FRU conditioning experiments larvae crawled on plain agar on a thermal gradient without the US present. In aversive NaCl conditioning experiments, larvae crawled on agar-NaCl gels on a thermal gradient (NaCl concentration 1.5 M) (Fig 2). Past work by other groups studying larva olfactory learning and memory made the aversive US present during testing ([10, 31, 32]), which ensured a clearer measure of associative memory, as sometimes conditioned movement is not observed or observed only weakly without the US present. We do the same in this paper.

The thermal gradient arenas (Fig 1) were used to measure the extent to which larvae approach or avoid the temperature used as the CS. Both thermal gradients used 20˚C and 27˚C as their coldest and warmest locations (left/right edges for linear gradient, center/perimeter for radial gradient), the same temperatures used as the CS during conditioning protocols. Larvae were initially spaced apart on a 5 cm vertical line in the center of the linear gradient, and placed on the perimeter of a 2.5 cm diameter circle on the radial gradient.

The efficiency with which larvae approach either extreme of a thermal gradient is extracted from raw trajectory data, along with many other parameters related to movement speed and turns. As noted above, the efficiency of movement along a gradient (the *x*-direction or *r*-direction) is quantified as a navigation index ($NI_x$ or $NI_r$), a dimensionless value with a possible range between −1 and + 1 (+ 1, for example, would be the navigation index if an entire population of larvae moved directly up the gradient without ever deviating). For reference, strong

linear chemical concentration gradients [51] yield *NI* near + 0.2, and strong thermal gradients [42, 43] yield *NI* near + 0.3, and both behaviors are robust in crawling larvae. The 20-to-27˚C gradients we use in this paper are strong (0.32˚C/cm for the linear gradient, 0.50˚C/cm for the radial gradient), but over a temperature range that does not normally elicit significant crawling in either direction (*NI* near zero, see [44]). Most conditioning protocols were tested on the radial gradient platform as well as the linear gradient. This allowed us to observe larvae navigate a more complex thermal landscape, and the circular geometry is well-suited for framing navigation as dispersal away from an initial temperature. For example, a larva crawling in a straight line will experience varying rates of temperature change as it moves.

## Results

### Thermal conditioning with fructose

Larva are capable of sensing extremely small changes in temperature [42, 43, 46, 55], but exhibit randomly directed exploratory motion [56] within a temperature range of approximately 22–28˚C, even in the presence of a spatial gradient [44]. We sought to test whether larvae form associative memories to temperatures within the normally neutral range when those temperatures are paired with a positive (fructose, FRU) or negative (salt, NaCl) tastant stimulus. If the association is established through conditioning, then their essentially random exploratory motion should change to include directed navigation towards (away from) the positive (negative) unconditioned stimulus.

As a first test of this approach, wild type $w^{1118}$ larvae were conditioned with FRU (2 M concentration) paired with 27˚C (Fig 2), placed on a spatial temperature gradient (Fig 1), and their locomotion was recorded with a camera and trajectories and other information extracted with software (see Materials and methods). In our experiments naive control larvae were moved between agar plates, alternating between room temperature and either 20˚C or 27˚C, but without any temperature-odor pairing (Fig 2B). These control larvae showed only very small (and statistically insignificant) net navigation on a gradient centered at 23.5˚C, as seen in Fig 3A (gray horizontal bars). Naive larvae navigated in a similar manner regardless of which temperature (27˚C or 20˚C) they were exposed to prior to testing. Larvae exposed to FRU-27˚C pairing moved up the gradient toward the warmer temperature, considerably more strongly (navigation index $NI_x$ = + 0.15) than the naive controls ($NI_x$ = + 0.03) (Fig 3A). Significant navigation toward the CS was also observed when larvae were conditioned with FRU paired with the cooler 20˚C, demonstrating that both the warm and cool ends of the neutral temperature range work for establishing associative learning.

Individual animals were tracked throughout every experiment, so we can also observe the distribution of navigation indexes within the population. Fig 3B shows histograms for control groups and both FRU-temperature pairings, and highlights the high level of variance in conditioned larva thermotaxis.

By looking more closely at individual turning events, we can also determine whether positive thermotaxis for larvae conditioned, for example, with the FRU-27˚C pairing are specifically attracted towards the warmer temperature. Larvae that increase turning rate during cooling and larvae that decrease turning rate during warming would both exhibit positive thermotaxis and move up the gradient, and the $NI_x$ summary metric would be unable to distinguish the two cases. By sorting all "run" segments (periods of straight crawling between turns) by the $dT/dt$, the rate of warming or cooling, that larvae experience, we can determine whether conditioned thermotaxis is an avoidant or attractive behavior (Fig 3C). With data framed this way, we can see that FRU-27˚C-conditioned thermotaxis is an *attractive* behavior, where larvae significantly reduce their turning rate when they experience the strongest warming,

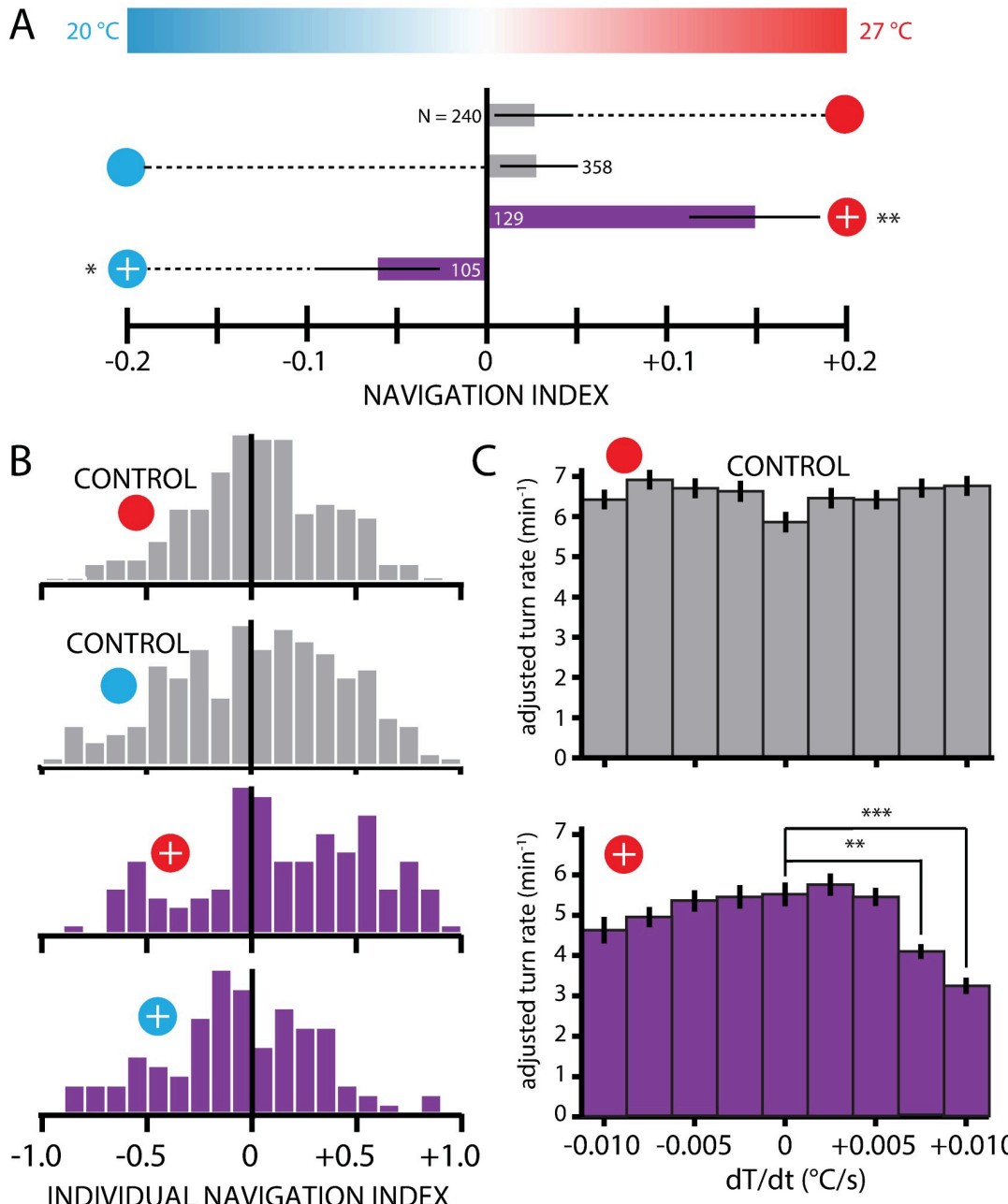

**Fig 3. Larvae form appetitive associative memory with temperature and FRU.** Thermotaxis measurements following paired FRU-27°C or FRU-20°C conditioning of $w^{1118}$ larvae. A: Net larval movement, summarized by the navigation index $NI_x = \langle v_x \rangle / \langle v \rangle$, on a linear temperature gradient (0.32°C/cm) from 20–27°C. A positive $NI$ indicates navigation towards the warmer temperature and a negative $NI$ navigation towards the cooler temperature, and magnitude indicates the strength of the navigation. Circle symbols connected to each result indicate the conditioning used, with the scheme from Fig 2. The number of larvae tested is indicated next to or inside of each bar. Appetitive conditioning with FRU induced significant navigation towards the conditioned temperature, for conditioning to 27°C and to 20°C. Significance tests were conducted with respect to the control (gray bars) group exposed to the matching temperature prior to testing. B: Histogram of the $NI$ values for individual larvae, for the same four experiment types in A. C. Turn rate as a function of the temperature change $dT/dt$ leading up to a turn, with crawling runs sorted into $dT/dt$ bins. Control larvae (top, gray bars) showed no significant difference between the $dT/dt = 0$ bin and any of the other eight bins. FRU-27°C conditioned larvae (bottom, purple bars) showed a strong drop in turn rate at the highest warming $dT/dt$ bins compared to the $dT/dt = 0$ bin. This indicates that the positive thermotaxis resulting from

conditioning is specifically an attractive behavior towards the warm temperature. The turning rates used here were adjusted to have crawling speed regressed out (speed is independent of crawling direction, see S1 Fig). Error bars are s.e.m. * indicates $p < 0.05$, ** indicates $p < 0.01$, *** indicates $p < 0.001$, Student's t-test.

whereas turn rates for all other $dT/dt$ groups are not significantly different compared to constant-temperature ($dT/dt = 0$) crawling. The turning rates used here were adjusted to have crawling speed (which is independent of crawling direction) regressed out, see S1 Fig.

Paired FRU (2 M) and 27°C conditioning was repeated with Canton-S larvae, another wild type strain. These larvae also navigated strongly up the gradient ($NI_x = + 0.18$), while the control larvae did not ($NI_x = −0.02$) (Fig 4). When a lower concentration of FRU (0.5 M) was used as a reward instead, Canton-S larvae were still able to move towards the warmer 27°C side of the gradient (Fig 4), but less robustly ($NI_x = + 0.06$) than when the conditioning was performed with 2 M fructose.

A loss of function mutant, $Ir25a^2$, missing a necessary co-receptor for primary cooling and warming sensation [46] was tested using the same FRU conditioning protocol (Fig 5). Conditioned $Ir25a^2$ larvae performed similarly to naive controls, whether FRU was paired with 27°C or with 20°C. This result implies that thermal associative learning involves sensory processing and is not merely a physical related to temperature and taste.

### Thermal conditioning with salt

We selected NaCl as the unconditioned stimulus (US) for aversive conditioning, as salt has been shown to be aversive at sufficiently high concentrations, and has been used to form associative memory in fly larvae when paired with odors [54, 57]. For all experiments with salt as a US, the agar gel for testing also included salt, at a spatially uniform concentration of 1.5 M, well above the threshold between attraction and aversion. Unexpectedly, non-conditioned

## FRU-TEMPERATURE CONDITIONING (Canton-S)

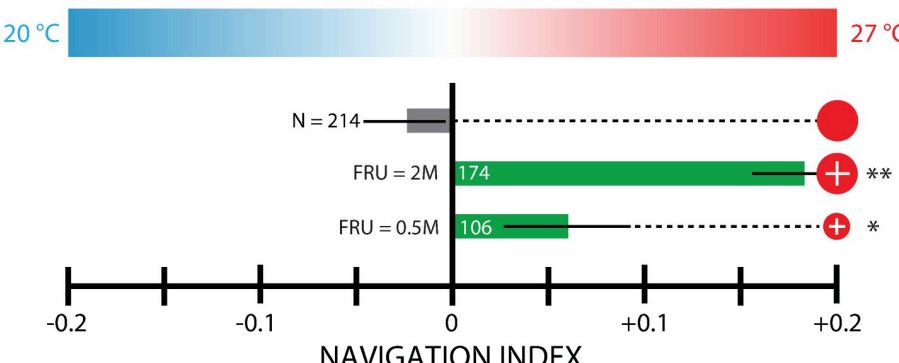

**Fig 4. Conditioned thermotaxis strength depends on conditioning concentration.** Thermotaxis measurements following paired FRU-27°C conditioning of Canton-S larvae, at two different FRU concentrations. The bar graph shows net larval movement, summarized by the navigation index $NI_x$, on a linear temperature gradient (0.32°C/cm) from 20–27°C. Circle symbols connected to each result indicate the conditioning used, with the scheme from Fig 2. The number of larvae tested is indicated next to or inside of each bar. Appetitive conditioning with FRU induced significant navigation towards the conditioned temperature, at both low (0.5 M, smaller red circle) and high (2 M, larger red circle) FRU concentrations, compared to non-conditioned control larvae. Larvae conditioned with the higher FRU concentration approached 27°C more strongly than the more weakly conditioned group. Error bars are s. e.m. Significance tests are with respect to the control (gray bar) group. * indicates $p < 0.05$, ** indicates $p < 0.01$, Student's t-test.

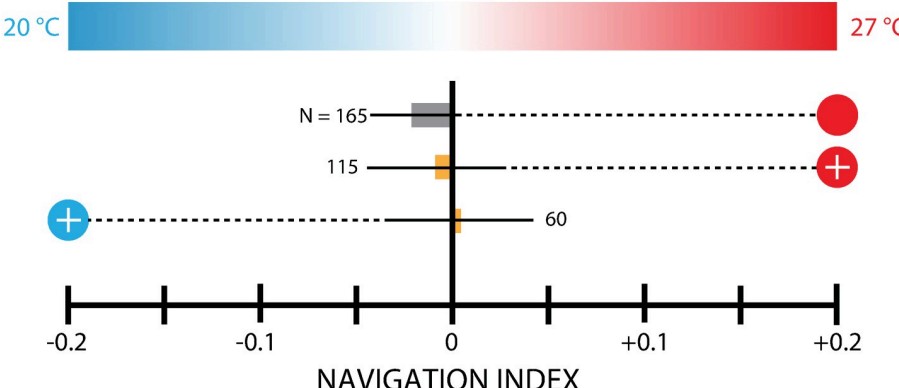

**Fig 5. Thermotaxis measurements following paired FRU-27˚C or FRU-27˚C conditioning of *Ir25a²* larvae, which are temperature-insensitive mutants.** Bar graph shows net larval movement, summarized by the navigation index $NI_x$, on a linear temperature gradient (0.32˚C/cm) from 20–27˚C. Circle symbols connected to each result indicate the conditioning used, with the scheme from Fig 2. The number of larvae tested is indicated next to each bar. Conditioned *Ir25a²* larvae performed no different than control when FRU was paired with 27˚C, or when FRU was paired with 20˚C. Error bars are s.e.m.

$w^{1118}$ control larvae placed on this NaCl agar gel actively crawled down the linear thermal gradient toward lower temperature, with a navigation index $NI_x = -0.12$ (when exposed to 27˚C) or $NI_x = -0.13$ (when exposed to 20˚C) (Fig 6A). We suspect this behavior may be due to a temperature dependence at the sensory level, where salt stimulates sensory neurons less strongly at lower temperature, so larvae perceive movement to lower temperature as movement towards lower salt concentrations (see Discussion). Regardless of the underlying reason for this non-conditioned thermotaxis, we should compare any result involving salt conditioning to the $NI_x = -0.12 or -0.13$ controls rather than to neutral $NI_x = 0$ navigation.

As a first comparison, larvae were conditioned using FRU and then tested on a NaCl gel. The resulting navigation depended on the conditioning temperature. Larvae that received FRU-27˚C conditioning overcame the existing tendency for negative thermotaxis on a thermal gradient salt gel (Fig 6B). FRU-20˚C conditioned larvae did not navigate significantly differently than the naive control larvae.

To test whether NaCl could be used for aversive conditioning, we paired NaCl with 27˚C prior to testing on a salt gel. Larvae navigated towards the cold side of the gradient more strongly than the control group, suggesting that a negative association with the warmer temperature can *add* to the existing tendency to move down this gradient in the presence of salt. Similarly, larvae conditioned with NaCl paired with 20˚C navigated towards the cold side of the gradient also, but *less* strongly than control group, suggesting that an aversive association with the cooler temperature can *subtract* from the existing tendency to move down this gradient. These two results show that, despite the unexpected negative baseline thermotaxis, larvae treated with aversive NaCl conditioning do form negative associative memories with a thermal conditioned stimulus.

## Double conditioning experiments

We also conditioned larvae using both FRU and NaCl as the US, each paired with a different CS temperature (see protocol schematic in Fig 2D). After conditioning, larvae were placed on

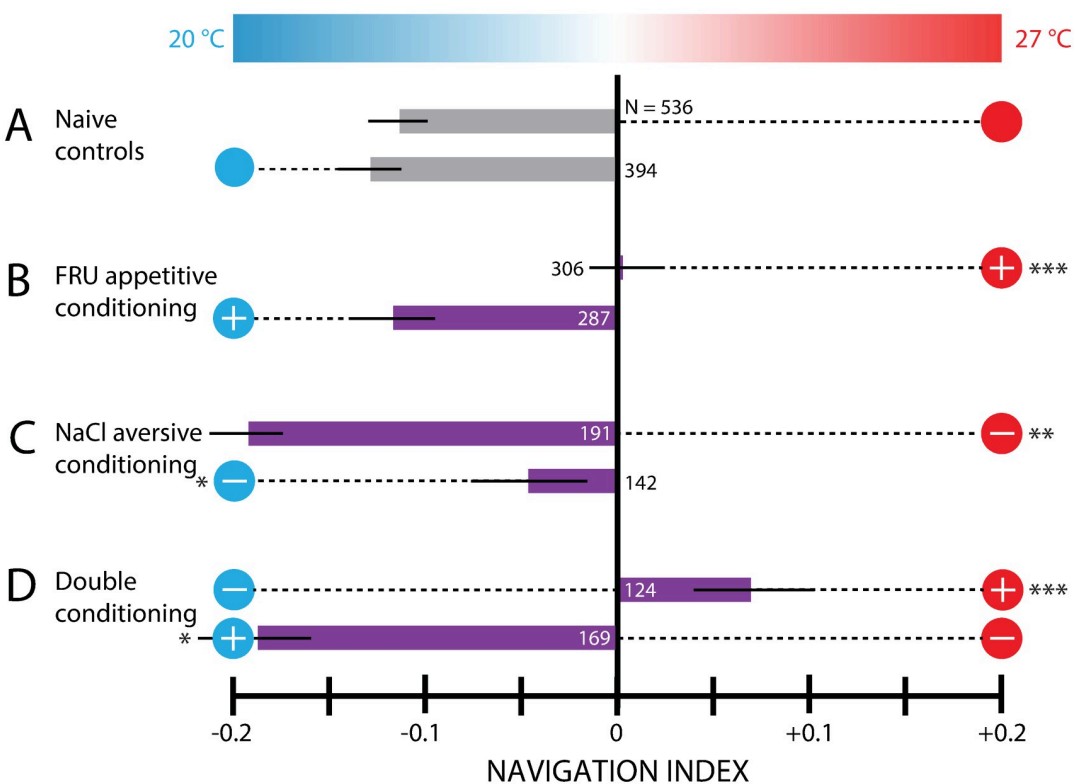

**Fig 6. Larvae form aversive associative memory with temperature and NaCl.** Thermotaxis measurements on an agar-salt gel following paired conditioning of $w^{1118}$ larvae. Bar graph shows net larval movement, summarized by the navigation index $NI_x$, on a linear temperature gradient (0.32˚C/cm) from 20–27˚C. Circle symbols connected to each result indicate the conditioning used, with the scheme from Fig 2. The number of larvae tested is indicated next to or inside of each bar. Experiments were performed on an agar gel with NaCl mixed in (1.5 M). A. Naive control larvae do not show neutral exploratory motion, but instead navigate towards the cool side of the gradient (27˚C, $NI_x = -0.12$; 20˚C, $NI_x = -0.13$). B. Larvae conditioned with FRU-27˚C and placed on a NaCl gel for testing do not exhibit net navigation, with the conditioning effectively canceling the tendency for the control group to move down the gradient. Conditioning with FRU-20˚C resulted in larval movement similar to that seen in control larvae. C. Larvae conditioned with NaCl-27˚C pairing move away from the warmer conditioned stimulus ($NI_x = -0.20$) more strongly than control group. Larvae conditioned with NaCl-20˚C pairing moved more weakly toward the cooler side of the gradient ($NI_x = -0.046$), less than the control group did. These results suggest that larvae form aversive associative memories with NaCl-temperature pairing, and perform thermotaxis accordingly, despite their natural tendency for negative thermotaxis on NaCl gel without conditioning. D. Larvae conditioned with FRU-27˚C and NaCl-20˚C moved up the gradient towards warmer temperatures, as opposed to both groups of naive control larvae, which move down the gradient in the presence of salt. Conditioning with the opposite scheme, FRU-20˚C and NaCl-27˚C, resulted in navigation down the gradient stronger than the 27˚C control group. These results indicate the double conditioning yields stronger conditioned thermotaxis, as the addition of FRU conditioning makes this graph essentially a heightened version of C. Error bars are s.e.m. Significance tests are with respect to the corresponding control (gray bar) group. * indicates $p < 0.05$, ** indicates $p < 0.01$, *** indicates $p < 0.001$, Student's t-test.

the linear thermal gradient. As with the single-conditioning using NaCl, salt was added to the gel in the testing arena (1.5 M). Therefore any navigation will be compared to the same negative-thermotaxing control groups exposed to 27˚C ($NI_x = -0.12$) or 20˚C ($NI_x = -0.13$). Since the two control groups do not show different thermotaxis, we compared to both groups together.

In one scheme larvae were conditioned with FRU paired with 27˚C, and NaCl paired with 20˚C. During testing, these larvae moved strongly towards the warmer side of the gradient,

whereas both control groups navigated towards the cold side, showing that a combination of appetitive conditioning to the warmer temperature *and* aversive conditioning to the cooler temperature can overcome and even reverse the tendency for larvae to move towards lower temperature in the presence of salt.

Conditioning with the opposite scheme, FRU paired with 20˚C and NaCl paired with 27˚C, resulted in strong navigation down the gradient, significantly stronger than the navigation of the control groups. This result demonstrates that the combined conditioning adds to the already existing tendency for negative thermotaxis on a thermal gradient salt gel.

Comparing Fig 6C and 6D, we observe that the additional conditioning with FRU enchances the effect of NaCl conditioning when fructose is paired with 27˚C.

## Linear gradient behavior with a counting index

Behavior was also quantified using a more traditional counting index method. Final larval location counts (left or right side of the starting position) for all FRU-temperature or NaCl-temperature (or both) conditioned larvae on the linear thermal gradient are shown in S2 Fig. This metric only relies on final locations and does not take into account the speed or efficiency of approach, but the results are consistent with the navigation index metric results (Figs 3–6), although the statistical significance levels tend to be less robust.

## Behavioral components and population distributions of conditioned thermotaxis

The navigation index (*NI*) as a summary metric of thermotaxis is sufficient to determine whether associative learning has taken place. But a more detailed look at the specific behavioral components that together determine the thermotaxis strength is warranted as well.

The table in S3 Fig shows the population histogram of $NI_x$ for all linear gradient experiments, along with how turning rate, turn size, and crawl speed depend on the crawling direction (up gradient, down gradient, or perpendicular to gradient). We also show turning direction and drift direction biases, and an efficiency metric that quantifies the degree of curvature during ostensibly straight runs.

Overall, we find a high variance in $NI_x$ among individuals, as in Fig 3B, and typically a difference in turn rate when larvae crawl up vs. down the thermal gradient. There is also a mild but consistent tendency to steer runs towards the warm side of the gradient, although in practice, given the generally high run efficiency, this would have only a mild effect on navigation compared to turn rate bias.

## Testing thermal conditioning with a radial gradient geometry

Larval navigation was also evaluated using a radial gradient (Fig 1B), with the same range of 20–27˚C, with the center held at one end of the range and the perimeter at the other. This geometry complicates the temperature stimulus inputs larvae experience during crawling. For example, a straight-crawling "run" segment of a larva's trajectory will always sense a constant $dT/dt$ temperature rate of change on a 1D *linear* gradient, but $dT/dt$ will vary during any run that is not parallel to a radius on a radial gradient.

Prior to navigation testing, wild type $w^{1118}$ larvae were conditioned with one of three single-pairings also used on the linear thermal gradient: FRU-27˚C, NaCl-27˚C, and NaCl-20˚C. All three conditioning protocols were tested on two radial gradient configurations: 27˚C inside and 20˚C inside (Fig 1B). Neither double conditioning experiments nor the FRU-27˚C pairing experiments were performed in this arena.

In the first gradient configuration, the perimeter is held at 27˚C and the center at 20˚C. Naive control larvae (with a plain agar testing gel) navigate away from the center with a radial navigation index of $NI_r = +0.20$ (Fig 7A). We note that because larvae start experiments near the center of the circular arena, all $NI_r$ values are positive, so any effect on thermotaxis due to paired conditioning will be relative to the positive control number, and not to $NI_r = 0$. Larvae conditioned with FRU-27˚C pairings navigated towards the warmer perimeter more strongly ($NI_r = +0.30$) than the control group (Fig 7A).

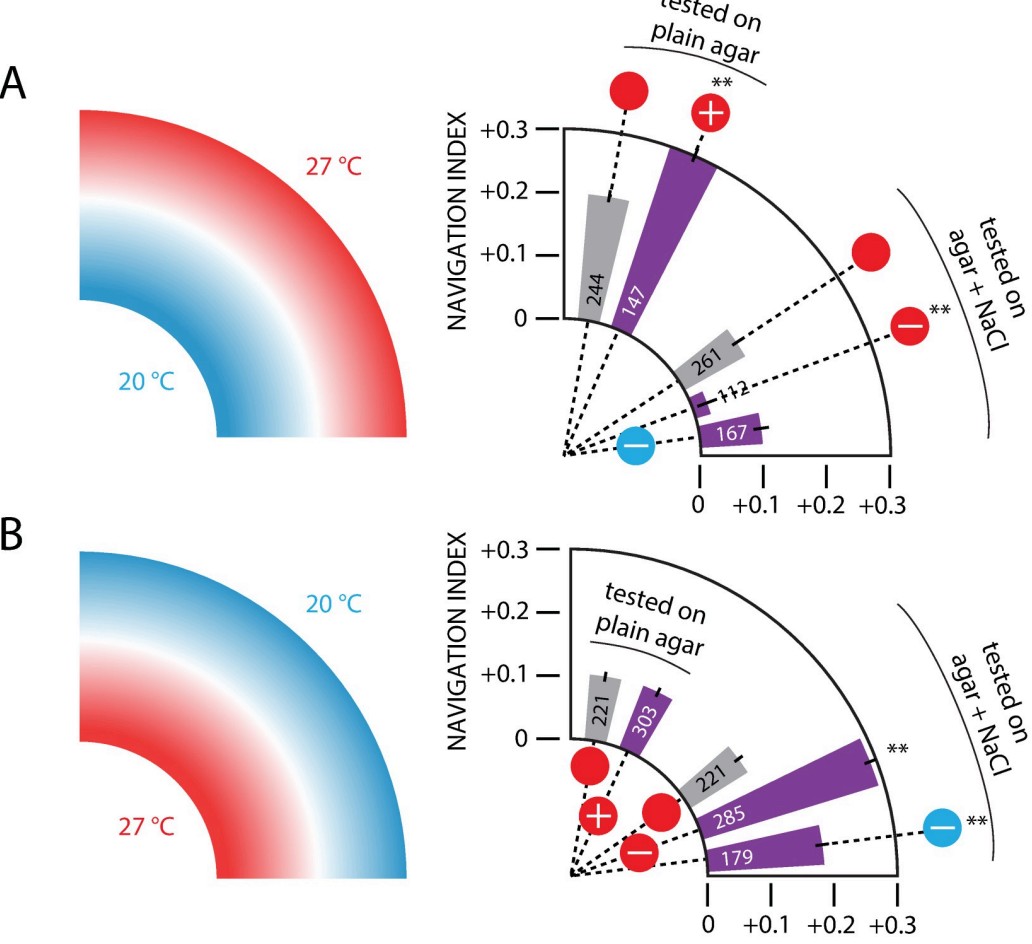

**Fig 7. Thermotaxis measurements in a radial gradient arena following single-paired conditioning in $w^{1118}$ larvae.** The gradient ranged from 20–27˚C, with a strength of 0.5˚C/cm. The polar bar graphs show net larval movement towards the perimeter, summarized by the radial navigation index $NI_r = \langle v_r \rangle / \langle v \rangle$. Circle symbols connected to each result indicate the conditioning used, with the scheme from Fig 2. Yellow boarders indicated experiments where NaCl was present in the testing gel. The number of larvae tested is drawn inside each bar. A: Radial gradient with 27˚C on the perimeter and 20˚C at the center. Larvae conditioned with FRU-27˚C pairing and tested on plain agar navigate to the outer edge more strongly than naive control larvae. Larvae conditioned with NaCl-27˚C pairing and tested on agar with 1.5 M salt navigate very weakly toward the outside compared to the control group, whereas larvae conditioned with NaCl-20˚C pairing navigate essentially the same as the control group. B: Reversed radial gradient with 20˚C on the perimeter and 27˚C at the center. Larvae conditioned with FRU-27˚C pairing and tested on plain agar navigate to the outer edge to the same degree as control larvae. Larvae conditioned with NaCl-27˚C pairing and tested on agar with 1.5 M salt navigate very strongly toward the outside compared to the control group, and larvae conditioned with NaCl-20˚C pairing navigate strongly to the outside as well. Error bars are s.e.m. Significance tests are with respect to the control (gray bar) group closest to the experiment group bar. ** indicates $p < 0.01$, Student's t-test.

Conditioning with NaCl prior to testing on this same warm-perimeter gradient leads to very weak outward navigation ($NI_r$ = + 0.03) with NaCl-27˚C pairing, but only slightly reduced (and not statistically different) navigation ($NI_r$ = + 0.10) with NaCl-27˚C pairing. These results are largely consistent with the equivalent linear gradient experiments (Fig 6). although here the NaCl-27˚C pairing result is more dramatic and the NaCl-20˚C pairing less significant. We suspect this is because larvae crawling in a radial thermal gradient geometry always initially travel in the + *r* direction, and will immediately encounter their aversive conditioned stimulus and adjust turning rate or other behaviors accordingly. When the conditioning is instead related to the center temperature, larvae will move away from it, and it may take several minutes of random exploration before they move towards the center and then bias their behavior. This time is a significant portion of the 10 minute experiment duration, so we expect conditioning effects on thermotaxis to be less noticeable when appetitive or aversive tastants are paired with the center temperature.

In the second gradient configuration, the perimeter is held at 20˚C and the center at 27˚C (Fig 7B). In this case, larvae conditioned with a FRU-27˚C pairing navigated similarly to the naive control group, paralleling the observations noted above regarding the NaCL-20˚C pairing. Both NaCl pairings (with 27˚C and with 20˚C) showed significantly altered $NI_r$ metrics. When conditioned to avoid the warm center, larvae moved outwards more efficiently than the control group, and did the same when conditioned to avoid the cooler perimeter.

Put together, these radial gradient experiments show larvae successfully traversing a more complex thermal landscape after associative conditioning.

## Discussion

### Larvae learn to approach temperatures paired with fructose

In this study, we showed that *Drosophila* larvae perform thermotaxis following a conditioning protocol that paired temperature with appetitive or aversive gustatory stimuli (Fig 3A). The observed change in behavior following conditioning is consistent with associative learning and shows that larvae can use temperature as a learning cue; to our knowledge this is the first time this has been done in this animal. The temperatures used as conditioned stimuli (27˚C and 20˚C) do not naturally elicit thermotaxis, and we therefore attribute the navigation behavior to associative learning from the conditioning. When fructose was paired with either the cooler or warmer temperature, larvae moved towards the appropriate conditioned stimulus.

Larvae conditioned with FRU paired with 27˚C dramatically reduced their turn rate when they experienced higher rates of warming during crawling (Fig 3C). Comparing to naive control larvae, which have an essentially constant turn rate for all warming and cooling levels, we conclude that the conditioned larvae not only move towards warmer temperature, but are in fact actively *attracted* to the conditioned stimulus. In other research studying fly larva navigation without conditioning, both positive and negative thermotaxis are characterized as inherently *avoidant* behaviors, where larvae increase turning rates when experiencing cooling below the neutral range (22–28˚C) or warming above the neutral range [42, 43, 45]. In the latter two articles, the equivalent of Fig 3C here is flat for positive $dT/dt$ but increases substantially on the left side. Our observations here notably show larvae using thermotaxis as an active seeking behavior, *reducing* turn rate to more efficiently move towards what they perceive is a better temperature, rather than increasing turn rate to avoid a worse temperature. We note that this method was not applied to the other experimental protocols due to a lack of statistical power.

Experiments with Canton-S larvae (Fig 4) establish that thermal associative learning takes place in at least one other genetic background, and that its strength depends on the tastant concentration, so that performance during testing improves with an increased intensity of the

reward during conditioning. Although the parameter space is too large to negotiate in the present work, future experiments could seek to fine-tune conditioning protocols and study more precisely the effects of altering conditioning doses or durations.

Overall these appetitive results are consistent with the model of associative memory of other stimuli in *Drosophila* [6, 9, 11, 58], and more generally across other taxa [3, 4, 15].

Conditioned *IR25a*[2] larvae showed no directed movement to conditioned temperatures (Fig 5), implying that associative learning does not occur in Ir25a[2] loss of function mutants. These experiments acted as a negative control, and establish that larvae are not primarily responding to other stimuli during the testing phase.

## Larvae navigate towards lower temperatures in the presence of salt

We observed that both groups of naive larvae tested on a linear thermal gradient moved toward the cooler side when a spatially uniform NaCl concentration (1.5 M) was present in the gel (Fig 6A). This result, while not expected, does not preclude using NaCl as an aversive US during conditioning experiments. Any thermotaxis measurement performed post-conditioning does need to be compared to the corresponding negative thermotaxis control result. For example, the double-conditioning experiment with NaCl-20°C pairing and FRU-27°C pairing shows that appetitive reward conditioning can overcome the negative thermotaxis bias and produce positive thermotaxis towards the rewarded warmer temperature (Fig 6D). Similarly, the opposite double conditioning scheme (FRU-20°C and NaCl-27°C) encourages increased larval thermotaxis toward the rewarded cooler temperature. Fructose conditioning alone was observed to overcome the innate thermotaxis bias when larvae experienced the FRU-27°C pairing prior to testing. However, this effect was notably stronger in the equivalent double conditioning protocol (NaCl-20°C pairing and FRU-27°C).

The shift in thermotaxis in the presence of salt that we have observed must be taken into account when designing future thermal memory experiments, and perhaps other aversive tastants without the shift would be preferable. The mechanism responsible for thermotaxis shift remains an open question, and uncovering the mechanism is beyond the scope of this paper. However, a possible explanation could be that the affinity of NaCl to gustatory receptors changes as a function of temperature. This would make it possible that the *perceived* NaCl concentration varies along the temperature gradient, even if the real salt concentration is spatially uniform. Previous observations noted that low concentrations of NaCl act as a reward in larval olfactory learning experiments [54, 59]. It should be possible in the future to characterize the sensitivity to NaCl at different temperatures with *in vivo* microscopy of gustatory sensory neurons expressing *Gr66a*, a known receptor for high salt concentrations like those used in our experiments here [60]. In larvae, *Gr66a* is expressed in the terminal organ ganglion (TOG), adjacent to the dorsal organ ganglion (DOG) that houses primary themosensory neurons [42, 45]. However, the TOG has also been reported to house thermosensory neurons [61], which could even be the same cells as the salt sensors. The proximity, or even overlap, of secondary thermosensors and salt sensors could explain the temperature dependence of salt sensation and therefore the anomalous thermotaxis in the presence of salt.

## Larvae learn to avoid temperatures paired with salt

Despite the unexpected thermotaxis in the presence of salt described above, larvae are still capable of forming associative memory with salt-temperature paired conditioning (Fig 6). When NaCl is paired with 27°C during conditioning, larvae crawl towards 20°C more strongly than the negative thermotaxing control group does ($p < 0.01$). Conversely for NaCl paired with 20°C, larvae move significantly less strongly towards 20°C ($p = 0.02$) as compared to the

corresponding control group. The second result shows that a negative association can be formed with the lower temperature, even when larvae normally move towards that temperature without conditioning.

Avoidance of 27˚C was also seen in larvae navigating the radial thermal gradient arena, in both orientations (Fig 7).

## Larvae perform associative-learning-based thermotaxis in complex thermal environments

Following conditioning, navigation on the radial gradient (Fig 7) is largely consistent with navigation on the linear gradient. Larvae were generally able to successfully navigate the radial gradient to seek out the temperature to which they have been conditioned regardless of valence. Specifically, the single conditioning pairing FRU-27˚C altered the navigation performance (as measured by $NI_r$) significantly when the CS temperature of 27˚C was on the perimeter of the circular arena, which matches the linear gradient results in Fig 3A. This did not happen when 27˚C was in the center (see possible explanation in the Results section). The single conditioning pairing NaCl-27˚C significantly altered the navigation performance in both configurations, again consistent with the linear gradient results from Fig 6C. The single conditioning pairing NaCl-20˚C was not consistent with the linear gradient results (no change in navigation with warm perimeter, change in the opposite direction for cool perimeter). We do not have a completely clear explanation for this, but note that the NaCl-20˚C paired conditioning with the linear gradient was the most subtle effect we observed (Fig 6C). The effect was statistically significant, but determining this required nearly 550 larvae combined between controls and aversive conditioning (the radial gradient equivalent had 300 larvae).

A specific advantage in using thermal stimuli for learning and memory experiments in larvae is that it is relatively straightforward to precisely manipulate the testing environment to create conditioned stimulus landscapes. This quality would allow for the design of more complex behavioral arenas to test the limits of larval thermal memory, and these arenas could be designed to specifically test extinction or replacement learning. Additionally, it would be of great value to expand these protocols to test different forms of memory including short-term, long-term, reversal learning, or operant conditioning using optogenetic rewards.

## Conclusion

The learned responses we have demonstrated are a result of the pairing of gustatory and thermal stimuli. The presence of conditioned thermotaxis behavior implies downstream neural circuitry where these sensory signals are integrated. Presumably, these signals converge in the mushroom body via a similar canonical circuit to that proposed in the current model for larval olfactory memory [28]. Single-cell-resolution neural imaging of larvae that are actively undergoing associative conditioning should be an exciting follow up to the thermal associative memory behavior experiments performed here. Investigation into candidate neurons involved in the pairing of thermal stimuli with other sensory inputs is warranted. The gustatory and olfactory sensory systems are highly interconnected, so it is not unreasonable to hypothesize that the gustatory fructose rewards and salt punishments used to establish thermal memory might utilize at least part of the established olfactory learning circuitry.

Our findings are also consistent with thermal learning experiments conducted in adult flies [49] suggesting that this behavior is present in both life stages. Previous studies in mushroom body neural circuitry have identified regions that undergo extensive structural changes as well as regions that appear relatively conserved in the adult [39]. In the future, it will be important

to identify GAL4 drivers that target neurons specific to this thermal memory behavior across life stages.

Overall, our experiments testing larval thermal memory indicate that larvae are capable of using neutral temperatures as a learning cue, with both positive and negative associations. We also demonstrate that larval thermotaxis can be turned into a active seeking behavior following appetitive and aversive conditioning. Thermal conditioning of *Drosophila* larvae could be adopted as an important tool to develop a more comprehensive understanding of learning, memory, and sensory integration.

## Supporting information

**S1 Fig. Slower larvae turn more frequently, independent of crawling direction.** Scatter plots from non-conditioned control larvae crawling on plain agar gel with a linear gradient (same experiments as Fig 3, topmost gray bar). Each individual "run" from all trajectories ($N = 240$ larvae) is considered separately, and its duration, average speed, and average crawling direction are extracted. Average turn rate is the inverse of the average run duration. A: Run duration vs. run speed, showing a significant correlation. A linear fit to the data provides a slope, which can then be used to regress the effect of speed out of run duration data. B: Run speed vs. crawling direction. These two quantities are not correlated, seen here for neutral thermotaxis crawling on a linear gradient, but also true when thermotaxis is present (see every entry in the "SPEED" column of S3 Fig).
(TIF)

**S2 Fig. Thermotaxis binary counting index analysis for larva conditioned with single or double tastant-temperature pairings.** This is an alternate method of determining preference, based solely on the final location of each larva after 10 minutes of activity on the linear gradient testing arena. The experiments are the same as used in Figs 3–6. Horizontal bars indicate the percentage of animals with final position to the LEFT ($N_L$) or RIGHT ($N_R$) of the starting location, and the number in the white rectangle for each experimental condition indicates the preference index, which is $(N_R - N_L)/(N_R + N_L)$, or equivalently, the difference between the right and left fractions. Significance tests are with respect to the control (gray bar) group for that strain and type of experiment. * indicates $p < 0.05$, ** indicates $p < 0.01$, *** indicates $p < 0.001$, Fisher's exact test.
(TIF)

**S3 Fig. Behavioral components of thermotaxis in larvae conditioned with single tastant-temperature pairings.** The experiments are the same as used in Figs 3–5, and used the linear thermal gradient for testing. Strain is indicated by a colored square: $w^{1118}$ (purple), Canton-S (green), and $Ir25a^2$ (orange). The conditioning protocol prior to thermotaxis testing is indicated by colored circles with + or − symbols as described in Fig 2. The navigation index $NI_x$ is the average of $NI_x$ for each individual larva, with the full distribution shown as a histogram and the average printed adjacent to it (red text indicating $NI_x > 0.04$ and blue text $NI_x < -0.04$). Turn rate, turn size, and speed are shown as a function of crawling direction, sorted into the wedges pictured above the columns (blue for the $-x$ direction, red for $+x$ direction, gray for $+/-y$ direction). Turn direction to $+X$ indicates the percentage of turns made following a run headed in the $+/-y$ directions that point the larva to the warm ($+x$) side of the gradient. Similarly, run drift to $+X$ indicates the percentage of runs that drift towards the warm ($+x$) side of the gradient. Finally, efficiency indicates how straight are the runs during the experiment set. For each run, efficiency is the ratio of the displacement to the path length, each animal's efficiency is the average of its run efficiencies, and the number in the table is the

average of all the animals' run efficiencies.
(TIF)

## Acknowledgments

The authors thank Bertram Gerber for useful discussions. We are grateful to James Baker and Sheyum Syed for fly support and advice, Julia Dallman for comments on the manuscript, Manuel Collazo for assistance with machining, and Golnoosh Goltapeh for assistance with experiments. We thank the University of Miami Biology and Physics Departments for internal support.

## Author Contributions

**Conceptualization:** Nikolaos T. Polizos, Mason Klein.

**Data curation:** Nikolaos T. Polizos, Stephanie Dancausse, Consuelo Rios, Mason Klein.

**Formal analysis:** Nikolaos T. Polizos, Mason Klein.

**Funding acquisition:** Mason Klein.

**Investigation:** Nikolaos T. Polizos, Stephanie Dancausse, Consuelo Rios.

**Methodology:** Nikolaos T. Polizos, Mason Klein.

**Project administration:** Nikolaos T. Polizos, Mason Klein.

**Resources:** Nikolaos T. Polizos, Mason Klein.

**Software:** Nikolaos T. Polizos, Stephanie Dancausse, Mason Klein.

**Supervision:** Mason Klein.

**Validation:** Nikolaos T. Polizos, Stephanie Dancausse, Consuelo Rios.

**Visualization:** Nikolaos T. Polizos, Mason Klein.

**Writing – original draft:** Nikolaos T. Polizos, Mason Klein.

**Writing – review & editing:** Nikolaos T. Polizos, Stephanie Dancausse, Consuelo Rios, Mason Klein.

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
