## [Decision Letter · Decision Letter 0]

7 Jun 2024

PONE-D-24-18023Drosophila larvae demonstrate associative learning and memory in response to thermal conditioningPLOS ONE

Dear Dr. Klein,

Thank you for submitting your manuscript to PLOS ONE. After careful consideration, we feel that it has merit but does not fully meet PLOS ONE’s publication criteria as it currently stands. Therefore, we invite you to submit a revised version of the manuscript that addresses the points raised during the review process.

We look forward to receiving your revised manuscript.

Kind regards,

Efthimios M. C. Skoulakis, PhD

Academic Editor

PLOS ONE

Journal Requirements:

Additional Editor Comments:

As you see in the appended comments there are significant questions and comments on methodology and data interpretation that need to be fully addressed before publication, as well as suggestions that should be taken into consideration in the revised manuscript. 

Reviewers' comments:

Reviewer's Responses to Questions

**Comments to the Author**

1. Is the manuscript technically sound, and do the data support the conclusions?

Reviewer #1: Partly

Reviewer #2: Partly

2. Has the statistical analysis been performed appropriately and rigorously? 

Reviewer #1: Yes

Reviewer #2: Yes

3. Have the authors made all data underlying the findings in their manuscript fully available?

Reviewer #1: Yes

Reviewer #2: Yes

4. Is the manuscript presented in an intelligible fashion and written in standard English?

Reviewer #1: Yes

Reviewer #2: Yes

5. Review Comments to the Author

Reviewer #1: The authors built a Drosophila larval learning paradigm to test the roles of temperature in associative learning, and they successfully demonstrated the associative learning using thermal cues as conditioned stimulus to pair with gustatory unconditioned stimulus. In addition, the author developed analysis pipelines for the locomotion trajectory data, especially for speed and turns, to depict the movement patterns of the larvae in learning and memory.

In general, there are several major questions the authors need to address before publication:

1. For the testing part, did the authors used all three arenas or just one of them for all experiments (appetitive, aversive, and double unconditioned stimulus association)? If they have the data for all three arenas, is there any difference between them? Is there anyone worked better, especially for the aversive association? If the authors could list this information clear, it would be helpful to understand their conclusions.

2. For the aversive associative learning part, especially Figure 7, personally I do not think the authors can draw the conclusion that “larvae treated with aversive NaCl Conditioning do form negative associative memories with a thermal conditioned stimulus”. The results need to show statistically significant difference between the trained and control groups. Indeed, the final number counts in S2 can be used to support the authors’ conclusion, and the radial gradient results really showed associated aversive learning. However, the authors may need to consider organizing the whole manuscript with a same testing method, the others can be used in supplementary data.

Maybe the low temperature reduces the salt sensation (for the control of aversive association), as the authors explained, so other aversive unconditioned stimulus may be taken into consideration. The authors also can try other training paradigms, such as electric shock or bitter taste solutions for unconditioned aversive stimulus, which are also commonly used.

3. Figure 6, the authors draw a conclusion “these results suggest that anosmic larvae are able to form association between tastant and temperature despite a loss of olfactory sensation”. The authors may re-phrase this sentence, as the statistics of this set of experiments is not significant difference.

4. For double unconditioned stimulus part, could this result reflect appetitive association with FRU plus the effect of NaCl during testing? The authors may consider another control group: FRU-Temperature conditioning, while testing on the NaCl agar with temperature gradients, then compare with the double unconditioned stimulus results.

There are some minor questions below:

1. Line 22 “due their to” should be “due to”.

2. In Figure 1, a legend is needed for the temperature heat map.

3. What do the traces in the inset of Figure 1 mean, are they representative locomotion traces, or the temperature measurement locations?

4. Line 231, “temperature-odor pairing” should be “temperature-taste pairing”.

In general, the authors first achieved to use temperature as a conditioned stimulus in Drosophila larval associative learning in their learning paradigm and arena, and they developed analytical tools and pipelines for learning behaviors. If the authors could provide more convincing results in the aversive learning part, or they can supply a unified/systematical testing and analysis for all learning paradigms (such as the radial gradient testing for appetitive, aversive, and double unconditioned stimulus learning), this work will be really outstanding in the field of Drosophila larval associative learning.

Reviewer #2: Review on „Drosophila larvae demonstrate associative learning.…“, by Polizos et al, submitted to PLOS ONE 2024

The authors report experiments in larval Drosophila melanogaster to test for associative learning between high/ low temperature as conditioned stimuli (CSs) and tastant reinforcers (or unconditioned stimuli, USs) (sugar or high concentration salt). I agree with the authors that temperature has some unique advantages as CS and that such a paradigm would be a very useful and significant addition to the available ‘toolbox’.

The main claim of the authors, reflected in the title, is that ASSOCIATIVE learning is demonstrated. This, I think, is a justified conclusion for the protocol shown in Figure 2C and the data gathered using this protocol in Figure 8. Here the authors show that when a relatively high temperature is paired with sugar and a relatively low temperature is paired with salt the larvae prefer the high temperature, whereas a preference for the low temperature is observed after reciprocal training (low temperature-sugar, high temperature-salt).

While this leaves open the question which of the four kinds of association indeed contributes how much to the eventual difference in temperature preference, this is convincing evidence that at least one association is formed.

This is convincing in particular because the two groups being compared are equated for all other parameters, such as temperature exposure, sugar- as well as salt-exposure, handling etc. By the same argument, however, the procedures shown in Figure 2B and the data gathered using that protocol are not convincing: no matter which comparison we look at, there is a confound. For example, considering the first versus the second protocol, any difference in temperature preference during the test could be due not to any association being formed, but to exposure to sugar versus exposure to salt per se (i.e. differences in satiation, osmotic stress, etc). Likewise, comparing the first to the third protocol, it might be that larvae that were kept at 27degrees for 3x5min for this reason alone have a different preferred temperature during the test as compared to larvae which were kept 3x5min at 20degrees (please do see my very last point below).

Given the present heading, the advice would be to restrict to that paradigm and data (but please do see my last point below). Whether that data, in w1118 mutants, is enough to warrant publication would be for the editor and the authors to decide. Personally, I think the scientific community deserves to know that the Figure2C/ Figure 8 paradigm works. If it were my study, I would consider adding wild-type data for that protocol, plus maybe the IR25a mutant and Orco mutant. But again, this is not for me to decide or even advise.

Title: Do the larvae demonstrate memory, or is it that the experiments demonstrate memory?

In the introduction in particular, references may better be restricted to the immediately relevant original papers plus a handful of pertinent reviews.

Please state more clearly what the room temperature during behavioural experiments was (state is only rearing temperature line 92); maybe add that also in Figure 2?!

As it is unconventional to perform the final temperature preference test in the presence of a US (for the case of salt), the authors should explain why they do so.

Maybe better to use either % or M when referring to concentrations?!

It is not entirely clear to me what the procedure for the “Control” was. Line 229 reads as if these were only handled, but the legend of Figure 2 reads as if they were exposed to the CSs (a “CS-only” control). If they were exposed to either 3x5min high or 3x5min low temperature, it would be interesting to see the data separated by these treatments. If there is no difference, you may show this and argue that, likely but not conclusively, the data in Figure 3 and related instances reflect associative memory…!

6. PLOS authors have the option to publish the peer review history of their article (what does this mean?). If published, this will include your full peer review and any attached files.

Reviewer #1: No

Reviewer #2: No

---

## [Author Response · Author response to Decision Letter 0]

22 Aug 2024

[We also uploaded a pdf of these responses in the Attach Files section]

RESPONSE TO REVIEWERS [PONE-D-24-18023] [Polizos, Dancausse, Rios, Klein]

(Original reviewer comments in brackets [[like this]], our responses in normal text)

REVIEWER #1:

[[The authors built a Drosophila larval learning paradigm to test the roles of temperature in associative learning, and they successfully demonstrated the associative learning using thermal cues as conditioned stimulus to pair with gustatory unconditioned stimulus. In addition, the author developed analysis pipelines for the locomotion trajectory data, especially for speed and turns, to depict the movement patterns of the larvae in learning and memory.

In general, there are several major questions the authors need to address before publication:

1. For the testing part, did the authors used all three arenas or just one of them for all experiments (appetitive, aversive, and double unconditioned stimulus association)? ]]

The linear gradient was our primary arena for experiments, as it is a simpler geometry and the stimulus experienced by the crawling larvae is more straightforward (e.g., the rate of change of temperature is constant for straight runs). We used the radial gradient arena for controls and three of the four single-conditioning protocols (FRU-27C, NaCl-20C, NaCl-27C), in both thermal geometries (warm inside/cold outside and cold inside/warm outside). This was previously stated in the second paragraph of the Radial Gradient subsection in Results, but we have added a direct statement about the types of experiments we did NOT perform on this setup. We have also adjusted Figure 2 (the protocol schematic figure) to emphasize the linear gradient as the testing arena, since every experiment type was tested there. 

[[If they have the data for all three arenas, is there any difference between them? ]]

For the conditioning protocols that were tested on all three arenas (FRU-27C, NaCl-20C, NaCl-27C):

(1) There was no difference for NaCl-27C

(2) FRU-27C was the same in the linear gradient and the radial gradient with 27C on the perimeter, but we did not observe the effect when 27C was in the center. A possible explanation for this was given already in the original submission (in the Results section), and has now been mentioned again in a greatly expanded section in Discussion as well. 

(3) NaCl-20C did not show aversive associative memory on the radial gradient. The aversive behavior is seen in the linear gradient, and is statistically significant, but is a more subtle effect, and we suspect was not observed in the radial gradient arenas due to a lower number of larvae tested and because the more complex geometry may mask more subtle navigation. 

All three of these comparisons are now more clearly listed and explained in the radial gradient subsection in Discussion. 

[[Is there anyone worked better, especially for the aversive association?]]

With new experiments added in this resubmission, establishing aversive association worked more clearly in the linear gradient, which both has a simpler geometry and a large number of new experiments performed in order to be statistically convincing. 

[[If the authors could list this information clear, it would be helpful to understand their conclusions.]]

Absolutely, and we thank the reviewer for asking about this. The differences between results in the various geometries are now extensively laid out in the Discussion section. 

[[2. For the aversive associative learning part, especially Figure 7, personally I do not think the authors can draw the conclusion that “larvae treated with aversive NaCl Conditioning do form negative associative memories with a thermal conditioned stimulus”. The results need to show statistically significant difference between the trained and control groups. Indeed, the final number counts in S2 can be used to support the authors’ conclusion, and the radial gradient results really showed associated aversive learning]]

[Note that the results in question are now part of Figure 6]. We agree with this assessment and in response have recorded a large number of new experiments, now including two types of control data (at the request of the other reviewer), and more experiments with the aversive NaCl conditioning. We now clearly show that aversive conditioning “works” in both directions, causing larvae to move away from 27C more strongly than controls (for NaCl-27C pairing) and away from 27C less strongly (for NaCl-20C). 

[[However, the authors may need to consider organizing the whole manuscript with a same testing method, the others can be used in supplementary data.]]

I believe this would no longer be necessary given the changes from the previous item? We note that we have revised the two supplementary figures (S2 and S3) to reflect all the new experiments. 

[[Maybe the low temperature reduces the salt sensation (for the control of aversive association), as the authors explained, so other aversive unconditioned stimulus may be taken into consideration. The authors also can try other training paradigms, such as electric shock or bitter taste solutions for unconditioned aversive stimulus, which are also commonly used.]]

This is an excellent point, and we actually tried both of these in our preliminary experiments before settling on NaCl (despite the unexpected negative thermotaxis in the control groups). We tried quinine as a bitter taste US, but could not reliably see a change in navigation in conditioned larvae. And we tried electric shock, which also produced inconclusive results, and had the added issue where electric current would heat the larvae, not ideal when temperature was our CS. It does seem important to mention this in the manuscript, so we have added sentences similar to what is written here to the section where NaCl protocols are first introduced in the Methods section. 

[[3. Figure 6, the authors draw a conclusion “these results suggest that anosmic larvae are able to form association between tastant and temperature despite a loss of olfactory sensation”. The authors may re-phrase this sentence, as the statistics of this set of experiments is not significant difference.]]

We agree, that statement was not accurate given the statistical significance we calculated. We ran many more experiments with hundreds more larvae since our initial submission, including additional controls, and the results are still inconclusive, if anything even less consistent. We have decided to remove the old Figure 6 and discussion of Orco2 mutant experiments. The result concerning anosmic larvae was quite secondary to the main conclusions of the manuscript, and we don’t have sound explanations for the results. The relationship between taste and smell sensing is likely more complicated than we first thought, and I think beyond the scope of this manuscript. 

[[4. For double unconditioned stimulus part, could this result reflect appetitive association with FRU plus the effect of NaCl during testing? The authors may consider another control group: FRU-Temperature conditioning, while testing on the NaCl agar with temperature gradients, then compare with the double unconditioned stimulus results.]]

This was a great suggestion, we are glad you provided it! We ran both FRU-27C and FRU-20C conditioning experiments, testing on an agar+salt gel. These can be seen in Figure 6B in the resubmitted manuscript. We used nearly 600 larvae total in these new experiments. The FRU-27C conditioning experiments moved towards the conditioned temperature more strongly than the corresponding control group did (also tested on the same salt gel). The FRU-20C group did not. 

The double conditioned larvae appear to navigate more strongly in the corresponding direction than the single FRU conditioned larvae, for both 20C and 27C conditioning, but neither is significant to within p<0.05 (the p-values are both <0.1, so they are probably different, just not with 95% confidence). 

In addition to the figure update, main body text has been adjusted to include the new experiments. 

[[There are some minor questions below:

1. Line 22 “due their to” should be “due to”.]]

Fixed, thank you.

[[2. In Figure 1, a legend is needed for the temperature heat map.]]

Temperature labels have been added to the heat maps for clarity, and colored scale bars have been added to both Temperature vs. position (x in panel A, r in panel B) graphs to make the scale more explicit. 

[[3. What do the traces in the inset of Figure 1 mean, are they representative locomotion traces, or the temperature measurement locations?]]

They are representative trajectories for the larvae. A sentence has been added to the Figure 1 caption to clarify. 

[[4. Line 231, “temperature-odor pairing” should be “temperature-taste pairing”.]]

This mistake has been removed while rearranging the sentences in this section. 

[[In general, the authors first achieved to use temperature as a conditioned stimulus in Drosophila larval associative learning in their learning paradigm and arena, and they developed analytical tools and pipelines for learning behaviors. 

If the authors could provide more convincing results in the aversive learning part, or they can supply a unified/systematical testing and analysis for all learning paradigms (such as the radial gradient testing for appetitive, aversive, and double unconditioned stimulus learning), this work will be really outstanding in the field of Drosophila larval associative learning.]]

We hope the new experiments we added are sufficient to meet this criterion. In particular, the aversive learning on the linear gradient should be much more convincing than in the original submission (see Figure 6A vs. Figure 6C). Larvae move less strongly towards (or more strongly away from) normally neutral temperatures following NaCl-temperature conditioning protocols. 

REVIEWER #2:

[[Review on „Drosophila larvae demonstrate associative learning.…“, by Polizos et al, submitted to PLOS ONE 2024

The authors report experiments in larval Drosophila melanogaster to test for associative learning between high/ low temperature as conditioned stimuli (CSs) and tastant reinforcers (or unconditioned stimuli, USs) (sugar or high concentration salt). I agree with the authors that temperature has some unique advantages as CS and that such a paradigm would be a very useful and significant addition to the available ‘toolbox’.

The main claim of the authors, reflected in the title, is that ASSOCIATIVE learning is demonstrated. This, I think, is a justified conclusion for the protocol shown in Figure 2C and the data gathered using this protocol in Figure 8. Here the authors show that when a relatively high temperature is paired with sugar and a relatively low temperature is paired with salt the larvae prefer the high temperature, whereas a preference for the low temperature is observed after reciprocal training (low temperature-sugar, high temperature-salt).

While this leaves open the question which of the four kinds of association indeed contributes how much to the eventual difference in temperature preference, this is convincing evidence that at least one association is formed.

This is convincing in particular because the two groups being compared are equated for all other parameters, such as temperature exposure, sugar- as well as salt-exposure, handling etc. 

By the same argument, however, the procedures shown in Figure 2B and the data gathered using that protocol are not convincing: no matter which comparison we look at, there is a confound. For example, considering the first versus the second protocol, any difference in temperature preference during the test could be due not to any association being formed, but to exposure to sugar versus exposure to salt per se (i.e. differences in satiation, osmotic stress, etc). 

Likewise, comparing the first to the third protocol, it might be that larvae that were kept at 27degrees for 3x5min for this reason alone have a different preferred temperature during the test as compared to larvae which were kept 3x5min at 20degrees (please do see my very last point below).]]

This will come up in the answers to several comments below, but our original submission did a poor job explaining what our controls were. The control group for the original submission was exposed to 27C (alternating with room temperature) prior to testing. And for this resubmission we added many experiments with a second control group that was exposed to 20C prior to testing. We believe this should address any concerns about the protocol comparisons, and we will elaborate in the responses below. 

[[Given the present heading, the advice would be to restrict to that paradigm and data (but please do see my last point below). Whether that data, in w1118 mutants, is enough to warrant publication would be for the editor and the authors to decide. 

Personally, I think the scientific community deserves to know that the Figure2C/ Figure 8 paradigm works. If it were my study, I would consider adding wild-type data for that protocol, plus maybe the IR25a mutant and Orco mutant. But again, this is not for me to decide or even advise.]]

Based on the additional experiments added to this submission (and more accurately describing the controls we already had), we believe the reviewer would agree that the old Figure2C/Figure8 (now Figure 2D/Figure6D) results are convincing, but so are the four single-conditioning experiment types, which all show altered navigation compared to the appropriate control (current Figure 3A and Figure 6C). 

[[Title: Do the larvae demonstrate memory, or is it that the experiments demonstrate memory?]]

A fair point! Given this mistake, and the more convincing results in the resubmission, we have changed the title of the manuscript. 

[[In the introduction in particular, references may better be restricted to the immediately relevant original papers plus a handful of pertinent reviews.]]

The large number of citations was there to help establish the state of the field and place our work in that context, but we agree there are probably too many. We have now removed the less relevant references, especially those related to mushroom body circuitry. 

[[Please state more clearly what the room temperature during behavioural experiments was (state is only rearing temperature line 92); maybe add that also in Figure 2?!]]

Thank you. This has been clarified (it’s approximately 24 C in our laboratory in Miami) several places, including in Figure 2A as suggested. 

[[As it is unconventional to perform the final temperature preference test in the presence of a US (for the case of salt), the authors should explain why they do so.]]

Also a very good point. It turns out this is fairly standard in fly larva learning experiments, and we used this method following advice from other researchers who primarily study larva learning (this is our first publication involving learning and memory). We have added an explanation in the Learning Protocols / “Testing” section of Methods, and included several citations where the researchers use this method. 

[[Maybe better to use either % or M when referring to concentrations?!]]

Thank you. We now only use wt./vol. % when initially referring to the agar and charcoal concentrations in making gels, as that seems to be the most common way to describe that in other larva papers. Everything else has now been converted to M. 

[[It is not entirely clear to me what the procedure for the “Control” was. 

Line 229 reads as if these were only handled, but the legend of Figure 2 reads as if they were exposed to the CSs (a “CS-only” control). 

If they were exposed to either 3x5min high or 3x5min low temperature, it would be interesting to see the data separated by these treatments. 

If there is no difference, you may show this and argue that, likely but not conclusively, the data in Figure 3 and related instances reflect associative memory…!]]

We apologize for our poorly explained controls, that was a bad mistake in the original submission. 

We have good news and more good news. The good news is that the control group was indeed exposed to the warmer temperature prior to te

---

## [Decision Letter · Decision Letter 1]

11 Sep 2024

Drosophila larvae form appetitive and aversive associative memory in response to thermal conditioning

PONE-D-24-18023R1

Dear Dr. Klein,

We’re pleased to inform you that your manuscript has been judged scientifically suitable for publication and will be formally accepted for publication once it meets all outstanding technical requirements.

Kind regards,

Efthimios M. C. Skoulakis, PhD

Academic Editor

PLOS ONE

Additional Editor Comments (optional):

Reviewers' comments:

Reviewer's Responses to Questions

**Comments to the Author**

1. If the authors have adequately addressed your comments raised in a previous round of review and you feel that this manuscript is now acceptable for publication, you may indicate that here to bypass the “Comments to the Author” section, enter your conflict of interest statement in the “Confidential to Editor” section, and submit your "Accept" recommendation.

Reviewer #1: All comments have been addressed

2. Is the manuscript technically sound, and do the data support the conclusions?

Reviewer #1: Yes

3. Has the statistical analysis been performed appropriately and rigorously? 

Reviewer #1: Yes

4. Have the authors made all data underlying the findings in their manuscript fully available?

Reviewer #1: Yes

5. Is the manuscript presented in an intelligible fashion and written in standard English?

Reviewer #1: Yes

6. Review Comments to the Author

Reviewer #1: The authors have made lots of changes according to the questions and suggestions.

1. The authors have made changes to clearly address their results according to the distinct testing platforms they used.

2. In general, I think the aversive conditioning results are quite convincing, especially in the linear gradient platform and the NaCl-27�C paring condition in of both radial gradient platforms. For the NaCl-20�C paring conditions, it showed aversive memory in the linear gradient platform, however, not in neither radial gradient platform. I think the authors need to be careful about their explanations here:

(1) Line 391, it should be “NaCl-20�C pairing” rather than “NaCl-27�C”.

(2) Line 400, the authors claimed that “conditioning effects on thermotaxis to be less noticeable when appetitive or aversive tastants are paired with the center temperature”. Indeed, when pairing NaCl with 27�C, larvae showed aversive memory in both radial gradient platforms. However, when pairing NaCl with 20�C, it did not show aversive memory in any radial gradient platform.

(3) Line 407, “When conditioned to avoid the warm center, larvae moved outwards more efficiently than the control group and did the same when conditioned to avoid the cooler perimeter”. I think the authors misinterpreted their data. Although the larvae from NaCl-20�C pairing in the gradient configuration with perimeter at 20�C and center at 27�C showed significance in navigation index compared to the naïve controls, it is attractive rather than repulsion to the 20�C.

The authors need to think about these results and give a possible explanation.

3. The authors supplied information about their trials using quinine or electric shock as aversive unconditioned stimulus. Personally, I think they can make it into a paragraph in the Discussion section instead of the Method section.

4. The author have added the results about testing the FRU-27�C and FRU20�C on an agar + salt gel plate, which makes their conclusion more accurate.

5. In addition, for the result part “Larvae navigate towards lower temperature in the presence of salt” (line 450), the authors can compare between Figure 3A and 6A, as well as the naïve control groups in Figure 7. May be these results could supply information about the differences between the platforms.

In sum, I think the authors have changed enough according to the comments, making their conclusions more accurate and convincing. Besides, the authors still need to make minor changes to improve the appearance of their work.

7. PLOS authors have the option to publish the peer review history of their article (what does this mean?). If published, this will include your full peer review and any attached files.

Reviewer #1: No

---

## [Editor Report · Acceptance letter]

14 Sep 2024

PONE-D-24-18023R1 

PLOS ONE

Dear Dr. Klein, 

I'm pleased to inform you that your manuscript has been deemed suitable for publication in PLOS ONE. Congratulations! Your manuscript is now being handed over to our production team.

Kind regards, 

on behalf of

Dr. Efthimios M. C. Skoulakis 

Academic Editor

PLOS ONE